# Addressing the Gap in Data Communication from Home Health Care to Primary Care during Care Transitions: Completeness of an Interoperability Data Standard

**DOI:** 10.3390/healthcare10071295

**Published:** 2022-07-13

**Authors:** Paulina Sockolow, Edgar Y. Chou, Subin Park

**Affiliations:** 1College of Nursing and Health Professions, Drexel University, Philadelphia, PA 19102, USA; sp3687@drexel.edu; 2Department of Medicine, Sidney Kimmel Medical College, Thomas Jefferson University, Philadelphia, PA 19107, USA; edgar.chou@jefferson.edu

**Keywords:** communication, home health care nursing, home health nursing, primary health care, continuity of patient care/standards, transition of care, nursing informatics, documentation

## Abstract

In a future where home health care is no longer an information silo, patient information will be communicated along transitions in care to improve care. Evidence-based practice in the United States supports home health care patients to see their primary care team within the first two weeks of hospital discharge to reduce rehospitalization risk. We sought to identify a parsimonious set of home health care data to be communicated to primary care for the post-hospitalization visit. Anticipating electronic dataset communication, we investigated the completeness of the international reference terminology, Logical Observation Identifiers Names and Codes (LOINC), for coverage of the data to be communicated. We conducted deductive qualitative analysis in three steps: (1) identify home health care data available for the visit by mapping home health care to the information needed for the visit; (2) reduce the resulting home health care data set to a parsimonious set clinicians wanted for the post-hospitalization visit by eliciting primary care clinician input; and (3) map the parsimonious dataset to LOINC and assess LOINC completeness. Our study reduced the number of standardized home health care assessment questions by 40% to a parsimonious set of 33 concepts that primary care team physicians wanted for the post-hospitalization visit. Findings indicate all home health care concepts in the parsimonious dataset mapped to the information needed for the post-hospitalization visit, and 84% of the home health care concepts mapped to a LOINC term. The results indicate data flow of parsimonious home health care dataset to primary care for the post-hospitalization visit is possible using existing LOINC codes, and would require adding some codes to LOINC for communication of a complete parsimonious data set.

## 1. Introduction

In a future where home health care is no longer an information silo, patient information will be communicated along transitions in care. Better information communication will improve patient care by reducing missed clinical opportunities and providing research data for hospitalization risk predictive analytics. Studies highlight shortcomings in patient information communication along transitions in care [1], especially to home health care and between home health care and physicians [2,3], which are impediments to care coordination [1]. This inadequacy occurs in the United States [4] (US) and abroad [5,6]. It exists despite studies indicating during transitions in care, effectual communication between clinicians is central to continuity of care [7,8,9,10] and sentinel events avoidance [11,12] for home health care patients.

A solution is electronic transmission of patient information [1,3,13] between electronic health record systems, and to embed home health care information into the routine outpatient clinical workflow for information to be more timely and accurate [1,3]. This communication necessitates implementation of reference terminologies (data standards) along the care transition.

Transition in care encounters with the primary care team occur for more than one million US patients, typically older than 65 years of age, annually discharged from hospital to home health care. The primary care team includes nurses and physicians. An evidence-based practice is that home health care patients have a primary care visit within two weeks of hospital discharge, [14,15] the Transition of Care visit. This practice has been shown to reduce mortality, cost of care [16,17], and rehospitalization risk in patients with sepsis [18] and heart failure [19]. There is no standardized Transition of Care instrument or data set to specify information needed for the visit.

Although primary care teams receive patient information from multiple sources, it is inadequate. Summary information from the hospital has an international data standard: the Continuity of Care document [20] which includes problems, medications, and procedures [4]. The Continuity of Care document, designed for physicians for information communication across care settings, Ref. [12] is not tailored to accommodate home health care clinical information [4]. Moreover, hospital communication presents patient information through an acute care lens instead of the broader community-based/ambulatory care perspective [21]. The latter would include the functional status of capabilities to address acute and chronic problems. Primary care teams may also receive paper hospital discharge information which has deficits. Missing information includes medication self-management capability, functional assessments [4], and problems [22]. Inaccurate information includes the medication list [23]. Information tends to be scattered, often across many pages of narrative text [22]. It is unknown whether the information hospitals send to primary care practices is sufficient for the Transition of Care.

In contrast, a rich set of structured data is captured during the home health care admission visit in the patient home. The home health care visit should occur within two days of hospital discharge, with patient assessment data collected in a standardized instrument, the Outcome and Assessment Information Set (OASIS) [24]. The OASIS contains 55 multiple-choice patient assessment questions [25], which include patient cognitive and functional capabilities and patient safety. A small subset of OASIS data is faxed to the primary care physician to request order sign-off for home health care reimbursement. This subset includes the reconciled medication list and proposed plan of care. The faxed form, received asynchronously relative to the Transition of Care visit, does not meaningfully engage physicians nor advance high-value care [3].

International standards for electronic communication of patient data exist, such as the Logical Observation Identifiers Names and Codes (LOINC) [26]. It is a reference terminology for identifying observations, health measurements, and documents for use in data exchange. For example, clinical observations are represented as variables, answer lists, and the groups which encompass them [27]. LOINC includes standardized patient assessment terms and has integrated several nursing terminologies [28,29], including OASIS [27]. LOINC is available from the LOINC website (http://loinc.org, accessed on 22 May 2022) at no cost.

Considering the absence of a standardized Transition of Care dataset, the deficits of hospital and home health care information communicated to primary care, and the richness of structured home health care data, we sought to identify a parsimonious set of OASIS data to be communicated from home health care for the Transition of Care visit. A parsimonious dataset would require minimal space on an electronic health record display screen and be easily viewable by clinicians. These characteristics improve the likelihood that access to this information would fit the Transition of Care visit clinical workflow. Furthermore, acknowledging that use of an international data standard is required for routine transmission of patient data along care transitions, we investigated the completeness of LOINC for coverage of the data to be communicated. Although electronic communication of home health care data to other care settings has yet to be studied, we anticipate electronic information flow along the transition in care will improve patient care.

## 2. Methods

Researchers conducted deductive qualitative analysis in three steps. They identified OASIS data available for the Transition of Care by mapping OASIS to the Transition of Care. To reduce the resulting OASIS data set to a parsimonious set clinicians wanted for the Transition of Care, researchers administered a survey to elicit Transition of Care clinician input. The team mapped the parsimonious dataset to LOINC and assessed LOINC completeness. The Drexel University and Jefferson University Institutional Review Boards approved the focus group and survey administration protocol.

### 2.1. Mapping of OASIS to Transition of Care

The researchers’ perspective was that of a primary care physician knowledgeable about OASIS data considering home health care information possibly needed by the care management team. The perspective was more comprehensive (e.g., medication self-administration issues) than that of a physician narrowly interpreting Transition of Care topics currently implemented (e.g., medication list). The broader perspective provided insight into the possible value of home health care data to the care management team.

The analysis used the current OASIS-D version [24]. The 55 patient assessment questions evaluated home safety, functional status, clinical status, and service needs [25]. OASIS data is structured: each question has an assigned code (e.g., M1000) with most question responses being categorical, usually with 3-to-6-point scales.

As there is not a nationally specified Transition of Care document, researchers used a primary care medical society Transition of Care [30]. It was comprised of information descriptions of measures—what needs to be addressed (referred herein as topics)—and did not have structured data fields. The study analysis focused on Transition of Care activities or decisions requiring data (e.g., Obtain and Review Discharge Information [17]). Transition of Care topics overlapped, for example, indicating review of similar data (e.g., Review Need for Follow-up on Pending Testing or Treatment; and Obtain and Review Discharge Information both include clinical status review). The Transition of Care incorporated nine clinical information topics distributed among three domains: functional status, clinical status, and service needs (e.g., Identify Available Community and Health Resources) (Table 1). Researchers excluded from the analysis a topic that did not refer to OASIS information due to asynchronicity (i.e., Communicate with Agencies and Community Services Used by the Beneficiary).

Researchers (EC, PS) independently extracted Transition of Care topics and OASIS questions and organized these on an Excel spreadsheet (Microsoft Corporation, Redmond, WA, USA), comparing their extractions and resolving differences. Together, they identified Transition of Care topics and OASIS questions that did not map or mapped between the Transition of Care and OASIS. OASIS questions which mapped to multiple Transitions of Care topics were noted. The analysis produced a list of topics and questions, each characterized as unmapped or mapped, and tagged with data source (Transition of Care, OASIS), domain, and unstructured/structured [25].

### 2.2. Identification of a Parsimonious OASIS Data Set for the Transition of Care Visit

The authors (EC, PS) elicited Transition of Care clinician input by developing and administering a survey asking what data was needed for a Transition of Care visit. Survey preparation involved: (1) development and testing of a focus group instrument with a reduced number of OASIS questions for the survey instrument; (2) draft survey instrument development; (3) focus groups to provide feedback on the draft survey instrument; and (4) survey instrument refinement, testing, and finalization.

Development of the focus group instrument involved the physician author (EC) applying his clinical experience to reduce the number of OASIS questions to a minimal subset needed to provide Transition of Care clinical information. Reduction involved four steps. First was removal of overlapping questions (e.g., retain ADL status questions, remove ADL goal questions which included status). Next, for over-arching questions with related granular questions, granular questions were removed (e.g., retain presence of deficit question, remove degree of deficit question). Third, clinically uninformative (e.g., number of therapy visits) or clinically unrelated (e.g., home safety, inoculation status) questions were removed. Retained questions were organized into groupings—questions that shared the same concept (e.g., the 5 medication-related questions are referred to as Medication Issues).

The retained 51 questions constituted 22 groupings which ranged from 1 question (15 groupings) to 9 questions (2 groupings). Most groupings contributed one question to the focus group instrument with a few groupings contributing multiple questions. The resulting instrument contained 33 items [25], each having a unique OASIS-related concept. The authors (EC, PS) tested the instrument separately with two primary care physicians to identify questions to be added or removed.

The same authors developed a draft survey instrument. Demographics questions asked: clinical role, age, years in practice, and electronic health record system used for the Transition of Care. The authors reduced the survey questions (stems) words to produce terse items designed for mobile device display. They developed two options for survey question responses. One option assumes the clinician always wants to see the data, whether there is a concern or not. Responses were: ‘Would include if yes’, ‘Would include if no’, ‘Not needed’. The simpler second option asked if the data was Needed, or Not Needed.

These authors conducted focus groups to provide feedback on the instructions, stems, responses, and survey format. Separate virtual focus groups were conducted for physicians and for nurses due to the expectation that responses would differ due to clinical role. They were recruited from the authors’ contacts and working groups in a national informatics professional association, AMIA (AMIA.org). Respondents were asked which stem questions to retain or exclude, how to simplify the question, and which response option was preferred.

The authors incorporated the feedback and optimized the survey design for handheld devices with a survey design expert (MS-see Acknowledgements). Survey administration was electronic with access via QR code (https://www.qr-code-generator.com/, accessed on 23 May 2022) and bitly URL (https://bitly.com/, accessed on 23 May 2022). The survey was distributed to the national AMIA informatics conference attendees at two presentation sessions and a working group meeting. Survey results identified OASIS concepts to be included in a parsimonious data set intended for communication to primary care teams for the Transition of Care visit.

### 2.3. Assess Completeness of LOINC

One author (SP) matched the parsimonious data set concepts to LOINC by mapping. This process used the Jupyter notebook environment (https://jupyter.org, accessed on 29 January 2022) to execute a custom python script. The script compared each LOINC term to OASIS concepts, output the matched LOINC terms and OASIS concepts, and exported the resulting table. SP compared the LOINC terms in the table to those on the LOINC website (https://search.loinc.org, accessed on 29 January 2022) using the “search” function to confirm the accuracy of the matches. Two authors (EC, PS) checked the LOINC mapping by consultation with two experts on LOINC, primary care, and gerontology data who are members of the U.S. Centers for Medicare and Medicaid Data Element Library Health IT Workgroup (TO, HM-see Acknowledgements).

The confirmed matches were graphically represented using visualization tools within Microsoft Power BI suite. Power BI includes a selection of visualization displays which could be created with different formats of input files (e.g., Excel, comma-separated, tab-separated) for computational languages (i.e., Data Analysis Expressions (DAX), Python, SQL, R). The Excel table of matches was uploaded to Power BI to create a stacked bar graph and a Sankey (network) diagram (Figure 1 is an example). The stacked bar displays the Transition of Care topics on the horizontal axis. For each Transition of Care topic are two vertical bars, each displays a count of OASIS and of LOINC matches The Sankey diagram depicts a more detailed analysis, showing connections between Transition of Care, OASIS, and LOINC (Figure 2 is an example). The latter are each a pillar of components: topic (Transition of Care), concept (OASIS) or term (LOINC). The diagram displays the links between the components.

The matching indicated completeness of the reference terminology relative to Transition of Care and OASIS. Incompleteness indicated areas for LOINC code recommendation development.

## 3. Results

### 3.1. Mapping of OASIS to Transition of Care

All 51 OASIS questions mapped to six Transition of Care topics, as shown in Table 2 and Figure 1. Often an OASIS question mapped to more than one Transition of Care topic, as shown in Table 2. Three Transition of Care service needs topics were unmapped (i.e., Interact with Other Clinicians who will Assume or Resume Care of the Patient’s System-specific Conditions, Assist in Scheduling Follow-up with Other Health Services, and Facilitate Access to Services Needed by the Patient and/or Caregivers) [25].

### 3.2. Identification of a Parsimonious OASIS Data Set for the Transition of Care Visit

#### 3.2.1. Focus Groups

Nurse focus group participants consisted of two nurse practitioners, a home health care nurse manager, and two nurse informaticians who are informal caregivers. They did not reach consensus on questions to retain. The nurse informaticians suggested adding questions to communicate patient baseline status to enable primary care clinicians to discern trends.

The physician focus group consisted of a general internist, a family practice physician, and a general internist in clinical informatics fellowship. They reached the following consensus:The question stem should communicate information available in the response choices;Data for all questions should be communicated regardless of whether the patient has a deficit. Therefore, the survey response should be whether the data is needed or not needed.

Neither focus group identified questions to be removed.

#### 3.2.2. Survey Administration

The survey was made available to approximately 75 conference attendees without regard for eligibility. Twelve physicians who cared for adult patients completed the survey as shown in Table 3. The typical survey respondent was a male physician over the age of 56 years with more than 20 years in clinical practice, who worked with a care coordination staff and used an Epic electronic health record system.

Almost all respondents chose to retain each of the 33 OASIS-related concepts (median 11, range 7). Respondents unanimously identified 10 OASIS-related concepts for retention (see Table 4). No concept had majority agreement for removal. When respondents identified a concept for removal, two reasons were selected: ‘too much information to incorporate in work process’ (8 of 9 respondents); and ‘prefer to collect this information directly with the patient’ (1 respondent). One respondent suggested adding a question about the need for a translator: information not captured in OASIS.

In response to the question, how would this information improve patient care, all respondents chose at least one of the multiple answers. The replies were: ‘reduce hospital use’ (11); ‘help identify management needs’ (10); and ‘reduce resource effort for contacting patients’ (8). Two respondents each wrote an additional response: ‘reduce need for duplicate testing’; and ‘reduce need to redocument/ask during the visit’.

Nine of the 12 respondents replied to the question asking why selected information would not be needed, selecting one or both answers. Eight responses included ‘too much information to incorporate in work process’. Three responses included, ‘prefer to collect this directly with the patient’.

Regarding where the home health care data should be displayed in the electronic health record system, all respondents replied. One response to the question was allowed. Responses included every choice with a consensus for a separate screen: ‘On a separate screen for homecare data’ (5); ‘On a snapshot screen’ (3); ‘Marked as homecare data on a screen with your practice’ (2); ‘On the screen with data from external sources’ (1); and ‘Unsure’ (1).

### 3.3. Assess Completeness of LOINC

The majority of OASIS questions in the parsimonious data set concepts mapped to a LOINC term (84%) as shown in Table 2 and Figure 2. This finding indicates data flow of OASIS survey question-related data to the Transition of Care is possible using existing LOINC codes, and would require adding some codes to LOINC for communication of a complete parsimonious data set.

## 4. Discussion

There is a persistent and widespread need for improved communication between home health care and primary care [2], especially during transitions of care [3]. This need is heightened during the two weeks following hospital discharge when a patient visit to the primary care team is associated with reduced hospitalizations [16]. Employing electronic communication of patient information between care settings is a solution whose time has come [13]. However, the data to communicate electronically are unspecified. We present the first study to investigate this gap. This paper adds to the literature evidence that a data standard was mostly sufficient to support the home healthcare to primary care communication of needed patient data. Our study reduced the number of standardized home health care assessment questions by 40% to a parsimonious set that primary care team physicians wanted for the post-hospitalization visit (Transition of Care). To characterize the potential to communicate the dataset electronically, our examination found the international LOINC reference data standard covered most (84%) of the concepts.

The finding that structured home health care data did not map to three Transition of Care service needs topics suggests an information deficit for Transition of Care visit decision-making (i.e., interact with other physicians who will assume/resume condition-specific care, assist in scheduling follow-up with other health services, and facilitate access to services needed by the patient and/or caregivers). Future research may identify other standardized data sources which could provide structured service needs data to the Transition of Care.

The parsimonious dataset of 33 items helpful for the Transition of Care almost completely incorporated the 23 hospitalization risk factors identified in Bick and Dowding’s [31] recent systematic review focused on home health care patients. Nine risk factors matched or incorporated (mapped to multiple items) 22 dataset items: living arrangements, caregiver, functional deficits, medication assistance, skin ulcer, dyspnea, urinary tract infection, psychological issues, and cognitive issues. An additional risk factor, recent hospitalization use, was not in the dataset as it was a condition of patient eligibility. Two risk factors were not among the dataset items, although they were OASIS-D questions (diabetes diagnosis, infusion therapy). Eleven risk factors were not OASIS-D admission questions (e.g., frequent recent provider visits, diagnoses) and not in the dataset. Conversely, the dataset included 10 items that did not appear among the risk factors: vision, pain, bowel issues (2 items), clinically significant medication issues (2 items), and issues prior to current illness (4 items). Overlap with the systematic review indicates completeness of the parsimonious dataset for Transition of Care primary care information needs.

Communication of electronically formatted home health care data could address the ineffectiveness of faxing home health care information [3]. Physicians anticipated having the dataset would improve patient care by reducing subsequent hospital use and identifying care management needs. They also expected having the information would reduce inefficient resource use. Furthermore, paper-based communication is an impediment to electronic communication of information [13] across transitions in care, and the realization of the benefits of improved data accuracy and timeliness [1]. Facilitating primary care clinician access to the electronic data would entail embedding home health care data into routine clinical workflow via the electronic health record system. Respondents preferred display of data on a separate electronic health record system screen for home health care data.

The almost complete mapping of the parsimonious dataset to LOINC supports home health care agencies’ potential to use the reference terminology to electronically transmit patient assessment data to primary care in preparation for the Transition of Care. An obstacle in the US is home health care technical and financial resource constraints. This challenge could be addressed with federal financial incentives similar to those for hospital and physician office electronic health record system adoption.

A study limitation is that the data source, Transition of Care guidance, had information topics broader than data elements, which required interpretation by the authors. Future work would incorporate Transition of Cares from diverse settings in the analysis to improve Transition of Care data element specificity. Moreover, considering the number of survey respondents, further investigation is warranted to assess whether the parsimonious dataset is generalizable to a larger and more diverse group of primary care clinicians.

Incorporating home health care data into primary care electronic health record systems as structured, standardized data could enable presentation of information during the post-hospitalization visit at the right time and support use of data management tools for clinician decision-making. Improved transition of care data communication could benefit patients in addressing their individual needs, while mitigating hospital readmissions. Electronic transmission of structured home health care data to primary care would facilitate data analytics: decision support, predictive modeling, and machine learning. These capabilities could enable development of new insights, informing the data capture process, system-generated interpretations, and their presentation to the primary care team in improving patient outcomes. Future research is warranted to assess the feasibility of this recommendation, and the impact on primary care workflow and patient outcomes.

## Figures and Tables

**Figure 1 healthcare-10-01295-f001:**
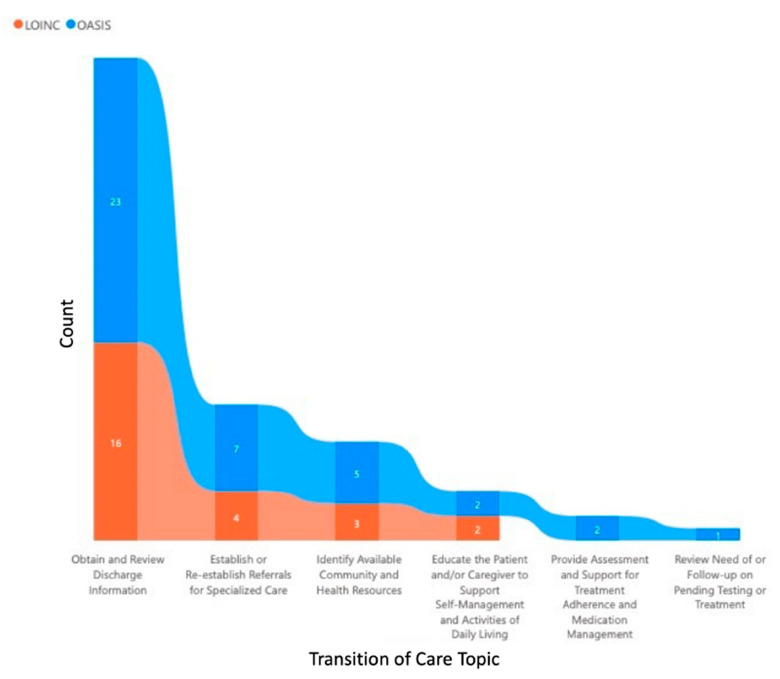
Stacked Bar Graph of Count of LOINC Codes and OASIS Concepts Mapped to Transition of Care Topics.

**Figure 2 healthcare-10-01295-f002:**
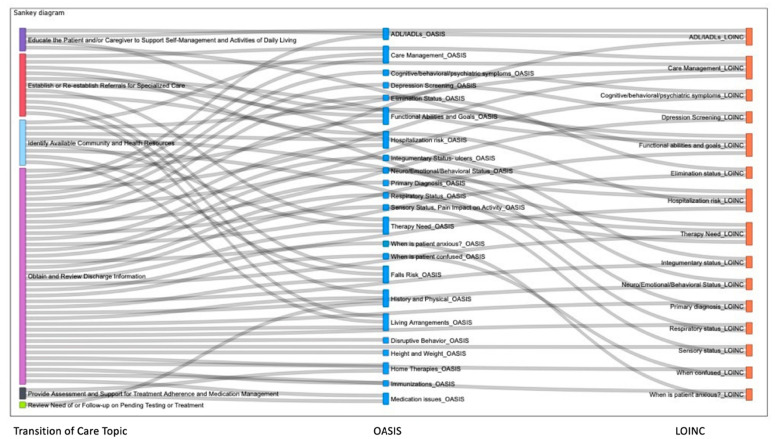
Sankey Diagram Illustrating Linkages Among Transition of Care Topics, OASIS Concepts, and LOINC Codes.

**Table 1 healthcare-10-01295-t001:** Transition of Care Topics’ Inclusion in Study Analysis.

Transition of Care Topic(s)	Domain	Included in Analysis
Obtain and Review Discharge Information	Clinical status	Yes
Establish or Re-establish Referrals for Specialized Care	Service needs	Yes
Educate the Patient and/or Caregiver to Support Self-Management and Activities of Daily Living	Service needs	Yes
Review Need for Follow-up on Pending Testing or Treatment	Clinical status	Yes
Identify Available Community and Health Resources	Service needs	Yes
Provide Assessment and Support for Treatment Adherence and Medication Management	Functional status	Yes
Assist in scheduling follow-up with other health services	Service needs	Yes
Communicate with a home health agency or other community service that the patient needs	Service needs	Yes
Facilitate access to services needed by the patient and/or caregivers	Service needs	Yes
Interact with other clinicians who will assume or resume care of the patient’s system-specific conditions	Clinical status	No

**Table 2 healthcare-10-01295-t002:** LOINC Coverage of the Transition of Care and Needed Home Health Care Patient Data.

Transition of Care Topic(s)	Patient Assessment Term(s) Related to OASIS, LOINC Codes	OASIS Question Code	LOINC Code
Obtain and Review Discharge Information; Establish or Re-establish Referrals for Specialized Care; Educate the Patient and/or Caregiver to Support Self-Management and Activities of Daily Living	Functional Abilities and Goals: Prior functioning-everyday activities Prior device use Self-care Mobility	GG0100 GG0110 GG0130 GG0170	83239-4 83234-5 57245, 89383, 89387–89388, 89393, 89396–89397, 89400–89401, 89405–89407, 89409–89410 57244, 85926-7, 89375–89386, 89390, 89393-5, 89398–9, 89403, 89408, 89411–89421
Obtain and Review Discharge Information	Primary Diagnosis Other Diagnoses Active Diagnosis	M1021 M1023 M1028	85920, 86255, 88489 85950-4 No corresponding LOINC
Obtain and Review Discharge Information; Review Need for Follow-up on Pending Testing or Treatment; Establish or Re-establish Referrals for Specialized Care	History and Physical	M1100	85950-4
Obtain and Review Discharge Information	Height and Weight	M1060	54567-3
Obtain and Review Discharge Information; Establish or Re-establish Referrals for Specialized Care; Identify Available Community and Health Resources	Hospitalization risk	M1033	57319-6
Obtain and Review Discharge Information; Establish or Re-establish Referrals for Specialized Care; Identify Available Community and Health Resources	Living Arrangements	M1100	85950-4
Obtain and Review Discharge Information	Neuro/Emotional/Behavioral Status	M1700	46589-8
Obtain and Review Discharge Information	When is patient confused	M1710	58104-1
Obtain and Review Discharge Information	When is patient anxious	M1720	86495-9
Obtain and Review Discharge Information	Cognitive, Behavioral, and Psychiatric Symptoms	M1740	46473-5
Obtain and Review Discharge Information; Provide Assessment and Support for Treatment Adherence and Medication Management	Home Therapies	M1030	46466-9
Obtain and Review Discharge Information	Sensory Status, Pain Impact on Activity: Vision Pain frequency	M1200 M1242	57215-6 No corresponding LOINC
Obtain and Review Discharge Information	Integumentary Status-ulcers: Presence of Stage 2+ pressure ulcer Number of unhealed pressure ulcers at each stage Number Stage 1 pressure ulcers Stage of most problematic pressure ulcer Presence of stasis ulcer Number of stasis ulcers Status of most problematic stasis ulcer Presence of surgical wound Status of most problematic surgical wound	M1306 M1311 M1322 M1324 M1330 M1332 M1334 M1340 M1342	85918-1 88494-0 No corresponding LOINC No corresponding LOINC No corresponding LOINC No corresponding LOINC No corresponding LOINC No corresponding LOINC No corresponding LOINC
Obtain and Review Discharge Information	Respiratory Status	M1400	57237-0
Obtain and Review Discharge Information	Elimination Status: Treated for urinary tract infection past 14 days Urinary incontinence or urinary catheter present Bowel incontinence frequency Ostomy	M1600 M1610 M1620 M1630	46552-6 46553-4 46587-2 86471-0
Obtain and Review Discharge Information	Depression Screening	M1730	57242-0
Obtain and Review Discharge Information	Disruptive Behavior	M1745	46592-2
Obtain and Review Discharge Information; Educate the Patient and/or Caregiver to Support Self Management and Activities of Daily Living	ADLs: Grooming Dress upper body Dress lower body Bathe Toilet transfer Toileting Transferring Ambulation Feeding	M1800 M1810 M1820 M1830 M1840 M1845 M1850 M1860 M1870	46595-5 46597-1 46599-7 57243-8 57244-6 57245-3 57246-1 57247-9 57248-7
Obtain and Review Discharge Information; Establish or Re-establish Referrals for Specialized Care; Identify Available Community and Health Resources	Falls Risk	M1910	57254-5
Obtain and Review Discharge Information; Provide Assessment and Support for Treatment Adherence and Medication Management	Medication Issues: Drug regimen review Medication follow-up Medication intervention Oral medication management Injectable medication management	M2001 M2003 M2005 M2020 M2030	57255-2 57281-8 57256-0 57285-9 57284-2
Obtain and Review Discharge Information; Establish or Re-establish Referrals for Specialized Care; Identify Available Community and Health Resources	Care Management	M2102	88465-0
Obtain and Review Discharge Information; Establish or Re-establish Referrals for Specialized Care; Identify Available Community and Health Resources	Therapy Need	M2200	57268-5

**Table 3 healthcare-10-01295-t003:** Demographics of Respondents who completed the Survey (N = 12).

	N
Age	
<40	3
40–55	3
56–75	6
Sex	
Female	3
Male	10
Clinical Role	
Nurse	0
Physician	12
Years in clinical practice	
<5	2
5–10	1
11–20	2
>20	7
Work with care coordination team	
Yes	9
No	3
Electronic health record system used in primary care	
EPIC	10
Cerner	1
e-ClinicalWorks	1

**Table 4 healthcare-10-01295-t004:** Survey Responses indicating OASIS Items needed for Transition of Care visit (N = 12).

Concept	# Responses = Needed	# Responses = Not Needed
Hospital Risk Predictors (e.g., falls, multiple hospitalizations/ER visits, mental status decline)	12	0
Living Arrangement (alone; availability of assistance)	12	0
Vision impairment (e.g., cannot read medication labels)	11	1
If Pain experienced: frequency	10	2
Stasis ulcers presence, status	12	0
Dyspnea presence, severity	10	2
Urinary Tract Infection in past 14 days	8	4
Urinary Incontinence or catheter presence	11	1
Bowel Incontinence frequency	11	1
Bowel Ostomy status	11	1
Cognitive functioning (e.g., level of alertness, comprehension)	11	1
When Confused (e.g., on awakening, night only)	9	3
When Anxious (e.g., daily)	7	5
Depression screening (PHQ2)	8	4
Cognitive, Behavioral, Psychiatric Symptoms (e.g., impaired memory, inability to perform usual ADLs)	12	0
Disruptive Behavior Symptoms frequency of (e.g., once a month, daily)	10	2
Grooming ability (e.g., dependent on others)	10	2
Ability to dress	11	1
Toilet transferring (e.g., bedside commode, bedpan)	12	0
Toilet hygiene (e.g., dependent on others)	12	0
Transferring (bed to chair, bed bound)	12	0
Ambulation/Locomotion (e.g., requires cane, can wheel self in wheelchair)	11	1
Ability to Feed self (e.g., dependent on others)	12	0
Clinically significant medication issues?	12	0
Patient/Caregiver high-risk drug education needed	8	4
Oral medications: Able to prepare/take reliably/safely	12	0
Injectable medications: Able to prepare/take reliably/safely	10	2
Types/Sources of assistance at home	11	1
Self Care ability prior to current illness (e.g., needed help; dependent on others)	10	2
Mobility prior to current illness (e.g., needed help; dependent on others)	10	2
Stairs ability prior to current illness (e.g., needed help; dependent on others)	10	2
Functional cognition ability prior to current illness (e.g., needed help; dependent on others)	10	2
Device use prior to current illness (e.g., walker, wheelchair)	10	2

## Data Availability

Data is available from the corresponding author.

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
