# Peer review of "Addressing the Gap in Data Communication from Home Health Care to Primary Care during Care Transitions: Completeness of an Interoperability Data Standard"

_healthcare, 2022, doi:10.3390/healthcare10071295_

Round 1

Reviewer 1 Report

The authors must explore other related efforts that cover the same or similar domain of activity.

Subsequently, in qualitative terms, the authors must compare their efforts. The authors need to consider the high light the novelty of the proposal being made.

Author Response

The authors must explore other related efforts that cover the same or similar domain of activity.

            Thank you for this suggestion. We added a phrase to the last paragraph in the Introduction stating, “Although electronic communication of home health care data to other care settings has yet to be studied…” We also added a sentence to the first paragraph of the Discussion, “We present the first study to investigate this gap.”

Subsequently, in qualitative terms, the authors must compare their efforts. The authors need to consider the high light the novelty of the proposal being made.

            We agree that comparison of our efforts to related efforts is important. Accordingly, as there are no related efforts, we did compare our findings to a relevant study by Bicks and Dowling in the Discussion section.

Reviewer 2 Report

This is a useful study that may improve transition of care data communication with patients’ benefits. I have only a few minor comments, as follows:

-        1. Authors employ several acronyms that hinder reading. If possible, some acronyms should be reported in full to easy reading.

-        2. The level of detail shown in Table 2 is not useful for study purposes and makes reading challenging. I suggest to simply the modality of data presentation and reduce the amount of details.

Author Response

  1. Authors employ several acronyms that hinder reading. If possible, some acronyms should be reported in full to easy reading.

We appreciate the suggestion and concur that acronyms can be troublesome to readers who are unfamiliar with the specific abbreviations. Throughout the revised version we have spelled out the acronyms for: HHC, TOC, CCD, and EHR. We have not changed the acronyms of LOINC and OASIS, as these are industry abbreviations for long, multi-word terms.

-        2. The level of detail shown in Table 2 is not useful for study purposes and makes reading challenging. I suggest to simply the modality of data presentation and reduce the amount of details.

We agree Table 2 presents a detailed analysis of the findings. Simplified views of the same data are displayed in Figures 1 and 2.

Reviewer 3 Report

Thank you for your manuscript entitled "Gaps in Data Communication from Home Health Care to Pri-mary Care During Care Transitions: Completeness of an In-teroperability Data Standard": it's very interesting.

However, minor revisions will need, specifically: references should be performed according to the journal's rules and a minor English language revision shoud be performed.

Author Response

Thank you for your manuscript entitled "Gaps in Data Communication from Home Health Care to Primary Care During Care Transitions: Completeness of an Interoperability Data Standard": it's very interesting.

Thank you.

However, minor revisions will need, specifically: references should be performed according to the journal's rules and a minor English language revision should be performed.

We have reviewed the document with the feedback in mind, seeking examples but finding none. The references were formatted by EndNote in APA 7, as specified in the invitation to submit an article that the authors received.